# Development and Performance Evaluation of a High-Permeability and High-Bonding Fog-Sealing Adhesive Material

**DOI:** 10.3390/ma14133599

**Published:** 2021-06-28

**Authors:** Tian Tian, Yingjun Jiang, Jiangtao Fan, Yong Yi, Changqing Deng

**Affiliations:** Key Laboratory for Special Area Highway Engineering of Ministry of Education, Chang’an University, South 2nd Ring Rd., Middle Section, Xi’an 710064, China; JiangtaoFan@chd.edu.cn (J.F.); yyong@chd.edu.cn (Y.Y.); changqingdeng@chd.edu.cn (C.D.)

**Keywords:** road engineering, fog-sealing adhesive material, epoxy-emulsified asphalt, recommended formulation, technical performance, microstructure

## Abstract

Herein, the effects of the contents of emulsified asphalt, waterborne epoxy resin emulsion, and curing agent on the permeability, bond shear strength, water stability, and aging resistance of epoxy-emulsified asphalt were studied. A formulation of epoxy-emulsified asphalt as a fog-sealing adhesive material was recommended, and a comparison between the fabricated adhesive material and a traditional Chinese fog-sealing adhesive material was conducted to verify the technical performance of the new material. In addition, the strength formation mechanism of the epoxy-emulsified asphalt was revealed via microcosmic analysis. Results show that the curing agent content mainly affects the permeability of epoxy-emulsified asphalt, and the emulsified asphalt content significantly affects the bond shear strength, water stability, and aging resistance. Moreover, the ratio of waterborne epoxy resin emulsion to the curing agent (epoxy ratio) has a certain effect on the bond shear strength. In the recommended formulation (a high-permeability and high-bonding fog-sealing adhesive material, which can be referred to simply as HPBFA), emulsified asphalt accounts for 80% of the total mass of the mixture, and the epoxy ratio is 2:1–3:1. It can improve air permeability, bond shear strength, water stability and aging resistance. The HPBFA-cured material exhibits a continuous three-dimensional network structure, hydrophobic surface, and large contact angle. Furthermore, the initial thermal weight loss temperature of the HPBFA-cured material is significantly higher than the environmental aging temperature. Additionally, the maximum temperature decomposition range is 0–160 °C, indicating improved strength, wear resistance, permeability, and aging resistance of the material.

## 1. Introduction

The preventive maintenance technology of asphalt pavement is developing with the development of asphalt concrete technology [1,2]. At present, the main conservation measures include crack sealing, fog sealing, chip sealing, slurry sealing, milling, and resurfacing, local regeneration, reconstruction, and so on [3,4,5,6].

Fog sealing is an economical, environmentally friendly, convenient construction and fast recovery technology, and there are more scholars to carry out related research. Montgomery et al. applied fog sealing material to longitudinal joints and compared it with the treatment method of void reducing asphalt membranes and concluded that the performance of fog sealing treatment was better [7]. Nadeem et al. evaluated the effectiveness of aerosol and regenerative seals by evaluating surface friction and durability and optimized the dosage [8]. Li, Qureshi et al. suggested that fog sealing could reduce the friction coefficient of the road surface and that it would take some time for the friction of the back-road surface to return to the original level [9,10]. However, Song et al. believe that fog sealing may improve the durability of open-graded friction course (OGFC) and extend its service life [11]. Lin et al. found that the recycled sealing material effectively softened the aging asphalt binder and reduced the high temperature performance of hot mix asphalt [12]. However, fog sealing is mainly performed through spraying an emulsified asphalt-based adhesive material on the road surface to fill the gap in the pavement structure, and the road maintenance period can be extended [13,14]. The material generally comprises highly permeable and strongly adhesive special asphalt, which must exhibit good wear- and water-resistance properties Therefore, the key to fog sealing technology is the development of fog sealing materials.

At present, the commonly used materials of fog seal are emulsified asphalt and styrene-butadiene-styrene (SBS) block copolymer modified emulsified asphalt. Jeong Hyuk et al. compared the curing time and adhesion behavior of polymer modified emulsion and unmodified emulsion [15]. The results show that the former has higher emulsion curing rate and material retention rate. Readul Mohammad et al. found that fog sealing reduces hydraulic conductivity [16]. In addition, emulsified asphalt still has some problems that are difficult to solve in the fog seal, such as the aging of emulsion under the action of ultraviolet light, poor permeability, and water sensitivity [17,18]. Therefore, it is of great significance to explore innovative fog sealing materials. Epoxy resins as a new type of adhesive material have been extensively tested by researchers worldwide.

Waterborne epoxy resin shows strong bonding ability and good permeability and can be used as an emulsified asphalt modifier for improved adhesion between aggregates [19]. An indoor pull-out test shows that the bonding strength of the sand-containing adhesive material increases as the amount of waterborne epoxy resin increases [20]. Epoxy asphalt shows better adhesion and indirect tensile strength than ordinary asphalt and exhibits improved fluidity when the epoxy fraction is below 50% [21]. Incorporating an epoxy resin in a warm asphalt mixture improves asphalt permeability [22]. Emulsified asphalt was modified using a cold mixture containing various epoxy resins, and this modified asphalt showed enhanced high-temperature stability, low-temperature crack resistance, and structural strength [23]. The interlayer failure and behaviors of rubber–powder-modified asphalt, SBS-modified emulsified asphalt, and epoxy resin adhesion in the waterproof bonding layer were studied [24]. The shear strength properties of the epoxy-emulsified asphalt were tested, indicating that the shear strength decreased with an increase in temperature, owing to decreased adhesion [25]. Microscopic analysis of a composite system containing waterborne epoxy resin–cement-emulsified asphalt showed that the material demonstrated good flexibility, high-temperature stability, and high bond strength [26,27,28]. A new type of fog-sealing layer asphalt was prepared by modification and emulsification, and its performance was analyzed [29]. An infrared spectroscopy study showed that fog-sealing and ordinary asphalts experience the same aging process, and both aging mechanisms were consistent [30]. Coating the fog sealant on the top of the open-graded friction course pavement rejuvenates the existing aged asphalt binder [8]. Road-core samples were analyzed before and after the fog-sealing treatment, showing a decrease in the aging rate of the asphalt binder owing to the decreased void ratio. The fog-sealing layer reduced water and air permeabilities, thereby improving the waterproof performance of the road surface and reducing the aging sensitivity caused by the oxidation of the binder [31]. The design of an epoxy-emulsified asphalt mixture was studied based on a modified Marshall test method [32]. The high- and low-temperature road performances and water stability of an epoxy-emulsified asphalt mixture were evaluated by performing a comparative analysis between an asphalt mixture and SBS-modified asphalt mixture [33,34].

In summary, epoxy-emulsified asphalt shows better permeability and cohesiveness than ordinary and SBS-modified emulsified asphalt [35]. Recently, epoxy-emulsified asphalt has been extensively used in pavement interlayer bonding and cold filling materials; however, it has rarely been used as a fog-sealing adhesive material. Moreover, to date, no test methods or indexes for evaluating the performance of fog-sealing adhesive material have been proposed, inducing considerable uncertainty and randomness in the selection of materials for fog-sealing layers.

Therefore, in this study, the effects of the contents of emulsified asphalt, waterborne epoxy resin emulsion, and curing agent on the permeability, bond shear strength, water stability, and aging resistance of epoxy-emulsified asphalt were studied by improving the laboratory test method. A formulation of epoxy-emulsified asphalt as a fog-sealing adhesive material was proposed, and the technical performance of the prepared material was verified by comparing it with a traditional Chinese fog-sealing adhesive material in the laboratory. The strength formation mechanism of epoxy-emulsified asphalt was revealed via microcosmic analysis. The research results are of great significance for improving the service life of asphalt pavement and enhancing the pavement maintenance technology system.

## 2. Materials and Methods

### 2.1. Materials

#### 2.1.1. Emulsified Asphalt

Table 1 shows the performance of the self-developed emulsified asphalt technology. The technical requirements of BE-4 type emulsified asphalt are implemented by referring to the requirements of PC-2 type emulsified asphalt in the Technical Specification for Construction of Highway Asphalt Pavement (JTG F40-2004).

#### 2.1.2. Waterborne Epoxy Resin Emulsion

The technical properties of the waterborne epoxy resin emulsion used in this experiment are shown in Table 2. The hydrophilic–lipophilic balance (HLB) value of the waterborne epoxy resin emulsion was 14.9, and this emulsion showed good centrifugal stability.

Waterborne epoxy resin emulsion was prepared by phase inversion, and the stability of the emulsion was taken as the main evaluation basis. The main preparation steps are as follows: the epoxy resin was heated to a certain temperature, and then stirred evenly by a high-speed shear dispersion emulsifier; complex emulsifier was added, and then hot water was slowly added and stirred evenly. The mixture was emulsified at the emulsifying temperature of 70 °C for 30 min to obtain waterborne epoxy resin emulsion.

#### 2.1.3. Curing Agent

The curing agent used in the test was obtained from Xi′an Huize Road Material Co., Ltd., Xi’an, China. Table 3 shows the technical properties of the curing agent.

#### 2.1.4. Aggregate Gradations

AC-13 and SMA-13 asphalt mixtures were used to simulate the actual asphalt pavement surface, on which the fabricated fog-sealing adhesive material was sprayed. Figure 1 presents the gradation of the mixtures. The asphalt/aggregate ratios of AC-13 and SMA-13 were 4.8% and 5.0%, respectively.

### 2.2. Experimental Program

The epoxy-emulsified asphalt mainly comprised emulsified asphalt, waterborne epoxy resin emulsion, and curing agent (Figure 2). To improve the material shelf life, the aforementioned three components of the material were generally stored separately. Each component was mixed according to the formula at room temperature and stirred evenly with glass rod before use.

Related studies have shown that, when the asphalt content in the epoxy resin asphalt is less than 50%, the entire resin is the matrix, and the material is susceptible to tensile fracture [36]. When the asphalt content exceeds 50%, the overall performance of the material tends to be like that of the asphalt, and the material fracture resistance shows significant improvements. Therefore, 50% is considered as the critical asphalt content in the epoxy-emulsified asphalt.

In the test process, the emulsified asphalt contents were 50%, 60%, 70%, 80%, and 90% of the epoxy-emulsified asphalt mixture, and the ratios of waterborne epoxy resin emulsion to the curing agent (epoxy ratio) were 2:1, 3:1, and 4:1. Note that the emulsified asphalt and epoxy ratio were represented by A and E, respectively. For example, epoxy-emulsified asphalt A_80_(E_2_) indicates that the emulsified asphalt accounts for 80% of the total mass of the mixture, and the epoxy mass ratio is 2:1.

### 2.3. Test Methods

#### 2.3.1. Penetration Test

In the quantitative penetration test, the determination of the actual penetration depth of the emulsified asphalt was hindered by problems such as an uneven penetration interface and “gas resistance” phenomenon (Figure 3). For this reason, the permeability test method proposed by Song and Meng was modified and improved [37,38]. Figure 4 shows the improved penetration test device, which is suitable for measuring and comparing the permeability of different fog-sealing adhesive materials.

The test steps mainly included the following: (1) First, 6 g of liquid to be tested and 60 g of standard sand were prepared. (2) Then, 60 g of standard sand was rotated around the opening of the funnel to ensure uniform density of the standard sand in the glass tube. Then, the sand filling time was recorded. (3) A total of 6 g of the liquid to be tested was poured into the glass tube using a funnel. The time required for the liquid to drop on the standard sand surface was measured. Further, the time required for the liquid to permeate on the standard sand surface was measured, and the penetration of the liquid was observed. If the liquid did not continue to infiltrate within 1 min, this period was recorded as the penetration time *T* of the liquid to be measured in the standard sand. (4) After the penetration of the liquid to be tested was completed, the glass tube was lifted and the standard sand remaining on the screen was weighed. (5) The aforementioned steps were repeated for different liquids to be tested, and the sand filling time was consistent. (6) Based on Equations (1) and (2), the permeability velocity *V* of the liquid to be tested was calculated and its permeability was evaluated.
(1)ρ=0.0012T+1.2907
(2)$$V=\frac{60-m_{1}}{A\rho t}$$
where *T* is the standard sand filling time, *ρ* is the loose density of the standard sand after *T* s, *V* is the penetration velocity of the liquid to be measured, *m*_1_ is the mass of the remaining standard sand on the screen, *A* is the effective area of the glass tube, i.e., $A=\pi \times {\left(\frac{2.5}{2}\right)}^{2}=4.91\;{\mathrm{cm}}^{2}$, and *t* is the penetration time of the liquid to be measured in the standard sand.

#### 2.3.2. Bond Shear Strength Test

The authors of this study independently developed a pavement interlaminar material shear tester (Figure 5). The size of the cylindrical specimen was *h* 50 mm × Φ 105 mm. The test parameters were set as follows: normal stress 0.7 MPa, loading rate 10 mm/min, and test temperature 20 °C.

The test steps mainly include the following. (1) Formed asphalt mixture rut board specimens. (2) The cylindrical specimen of *h* 50 mm × Φ 100 mm was drilled on the rut board. (3) Two core specimens were prepared, and their upper (or lower) surfaces were coated with 0.5 kg/m^2^ of fog-sealing adhesive material. Then, they were placed in a curing box with a constant temperature of 20 °C for 1 h. (4) The surface of the two core specimens coated with the fog-sealing adhesive material was adhered, and the specimens were placed in a standard curing room for 7 days. (5) The cured fog-sealed adhesive specimen was placed in a shear mold, and the bonding surface of the specimen was placed on the same horizontal line as the sliding surface of the shear mold. (6) The assembled “specimen + mold” was placed on the sliding track of the shear tester and fixed with a baffle. Further, the test parameters were set, and the load column was placed in contact with the upper part of the specimen. (7) At the beginning of the test, the lateral load column moved horizontally to provide horizontal thrust for the side-by-step loading of the lower specimen. The bond shear strength $\tau_{f}$ of the specimen was calculated according to Equation (3).
(3)$$\tau_{f}=\frac{F}{S}$$
where *τ_f_* is the bond shear strength, *F* is the shear force (destructive load), and *S* is the force area, i.e., *S* = π × 50^2^ = 7870 mm^2^.

#### 2.3.3. Water Stability Test

The test steps mainly included the following. (1) An asphalt mixture cylindrical specimen with dimensions *h* 63.5 mm × Φ 101.6 mm was molded. (2) Then, 1.0 kg/m^2^ of fog-sealing adhesive material was weighed, evenly brushed on the entire specimen surface, and cured in the standard curing room for 3 days. (3) According to the Chinese standard JTG E20-2011, the residual stability MS_0_ and strength ratio TSR of the fog-sealed adhesive specimen after curing were examined.

#### 2.3.4. Aging Resistance Test

The antiyellowing aging box independently developed by Chang′an University was adopted as the test instrument (Figure 6). The test parameters (Chang′an University, Xi’an, China) were as follows: irradiation intensity 250 W/m^2^, aging temperature 60 °C, irradiation height 30 cm from the aging specimen, and aging time 48 h.

The test steps mainly include the following: (1) Mass *m*_0_ of the test mold was weighed. Then, the prepared epoxy-emulsified asphalt was poured in the test mold and placed in a thermostat below 40 °C for curing. When the specimen mass remained unchanged, mass *m*_1_ was recorded. (2) The cured specimen was placed in the aging box for photothermal aging for 48 h, and the mass of the aging specimen was *m*_2_. The mass-loss rate after photothermal aging was calculated using Equation (4).
(4)$$M=\frac{m_{1}-m_{0}}{m_{1}-m_{2}}$$
where *M* is the mass-loss rate after photothermal aging, *m*_0_ is the mass of the test mold, and *m*_1_ and *m*_2_ are the mass of the specimen before and after photothermal aging, respectively.

## 3. Results and Discussion

### 3.1. Fabrication of the Fog-Sealing Adhesive Material

#### 3.1.1. Permeability

Table 4 shows the permeability test results of the different tested liquids. There were three parallel samples in a group, and the results were representative values according to the Grubbs method.

Among the three components of the epoxy-emulsified asphalt, the curing agent almost does not penetrate the standard sand, and the difference in the penetration velocity of the waterborne epoxy resin emulsion and emulsified asphalt is very small, which is approximately 0.23 times that of water; moreover, the permeability is good. Therefore, the permeability of epoxy-emulsified asphalt is mainly affected by the curing agent content.

Figure 7 presents the penetration test results of the epoxy-emulsified asphalt with different curing agent contents.

Based on Figure 7, the permeability of epoxy-emulsified asphalt decreases exponentially with an increase in the curing agent content. When the curing agent content is 0–10%, the permeability of epoxy-emulsified asphalt sharply decreases with an increase in the curing agent content. In other words, the permeability decreases by approximately 16% when the curing agent content is increased by 1%. When the curing agent content is 10–15%, the decrease rate of the permeability velocity of epoxy-emulsified asphalt reduces, and the permeability velocity decreases by approximately 10% when the curing agent content is increased by 1%. When the curing agent content is >15%, the permeability does not decrease considerably.

Zhang et al. showed that, when the curing agent content reached approximately 10%, the epoxy reaction efficiency will be affected [39]. This will result in a low crosslinking density of curing materials, which could hinder the formation of an ideal crosslinking network, yielding low-permeable epoxy asphalt. Hence, the curing agent content in epoxy-emulsified asphalt should not exceed 10%.

#### 3.1.2. Bond Shear Strength

Figure 8 presents the test results of the bond shear strength of the specimens. There were six parallel samples in a group, and the results were representative values according to the Grubbs method.

Based on Figure 8, the epoxy-emulsified asphalt with an epoxy ratio of 3:1 exhibits the highest bond shear strength. This indicates that the curing agent content is related to the bond shear strength, and there is an optimal curing agent content. When the emulsified asphalt content is increased, the bond shear strength of epoxy-emulsified asphalt gradually decreases.

When the emulsified asphalt content is greater than 80%, the bond shear strength of epoxy-emulsified asphalt decreases rapidly. In other words, when the emulsified asphalt content is increased by 10%, the bond shear strength of specimens decreases by approximately 25%. This may be because, when the amount of emulsified asphalt is small, no effective oil film thickness is formed between layers, and the bond shear strength increases with the increase of the amount of emulsified asphalt. When the amount of emulsified asphalt begins to exceed a certain amount, the excess emulsified asphalt will form a sliding surface between layers, which will reduce the bond shear strength. When emulsified asphalt exceeds the optimal amount, a weak sliding surface is formed between layers, and the bond shear strength will decrease rapidly, which is consistent with the performance in this paper when the emulsified asphalt amount is >80% [40]. Therefore, the emulsified asphalt content should ideally not exceed 80%.

#### 3.1.3. Water Stability

Figure 9 depicts the water stability test results of the specimens. There were six parallel samples in a group, and the results were representative values according to the Grubbs method.

Figure 10 shows that the MS_0_ and TSR values obtained for the fog-sealing adhesive specimens with different component contents are improved compared with the control specimens. When the emulsified asphalt content is 80%, the degree of improvement is high. The residual stabilities of A_80_(E_2_), A_80_(E_3_), and A_80_(E_4_) are 97.7%, 96.4%, and 95.0%, respectively, while the freeze–thaw splitting strength ratios are 81.3%, 80.5%, and 79.6%, respectively. Compared with the control specimens, the MS_0_ values are increased by 10.6%, 9.2%, and 7.6%, and the TSR values are increased by 15.3%, 14.2%, and 12.9% in the case of A_80_(E_2_), A_80_(E_3_), and A_80_(E_4_), respectively. This result may be attributed to the gradual increase in the free asphalt in the mixture with increasing emulsified asphalt contents, filled parts of the original void surfaces, reduced numbers of voids in the mixture, and improved permeability and bonding properties of the mixture to a certain extent [41].

#### 3.1.4. Aging Resistance

Figure 10 presents the mass-loss rate of the specimen after the photothermal aging test. A lower mass-loss rate signifies improved aging resistance of epoxy-emulsified asphalt. There were six parallel samples in a group, and the results were representative values according to the Grubbs method.

Based on Figure 10, the mass-loss rate of epoxy-emulsified asphalt is as follows: epoxy ratios 3:1 > 2:1 > 4:1. When the emulsified asphalt content is increased, the mass-loss rate of epoxy-emulsified asphalt after photothermal aging gradually decreases. When the emulsified asphalt content is ≥ 80%, the mass-loss rate starts stabilizing, and no significant difference is observed in the mass-loss rate of epoxy-emulsified asphalt under the three epoxy ratios. In other words, the curing agent content has a slight effect on the mass-loss rate of epoxy-emulsified asphalt after photothermal aging. Therefore, the emulsified asphalt content should ideally not be <80%.

#### 3.1.5. Recommended Formula for Fog-Sealing Adhesive Material

The technical performance of epoxy-emulsified asphalt shows obvious differences under different component contents. Table 5 recommends composition contents for preparing epoxy-emulsified asphalt from the perspective of technical performance.

Thus, the recommended formulation of epoxy-emulsified asphalt for a fog-sealing adhesive material is 80% emulsified asphalt and 2:1–3:1 epoxy ratio. In the subsequent performance evaluation, the experimental study adopted epoxy-emulsified asphalt A_80_(E_2_) and named this high-permeability and high-bonding fog-sealing adhesive material as HPBFA.

### 3.2. Evaluation of the Fog-Sealing Adhesive Material

#### 3.2.1. Permeability

The penetration test results of HPBFA and other materials are shown in Table 6. There were three parallel samples in a group, and the results were representative values according to the Grubbs method.

Table 6 shows that the penetration velocity of HPBFA is 1.29 cm/min, which is 0.54, 8.06, and 3.69 times higher than those of BE-4-emulsified asphalt, SBS-modified emulsified asphalt, and the asphalt commonly used in fog-sealing layers, respectively. The results show that the permeability of HPBFA is slightly lower than that of BE-4-emulsified asphalt but still exhibits obvious advantages.

Figure 11 displays the water-permeability coefficients of the specimens after brushing the fog-sealing adhesive material with different contents. There were three parallel samples in a group, and the results were representative values according to the Grubbs method.

Figure 11 shows that, when the amount of the different applied types of the fog-sealing adhesive material was increased, the changing trend of the water-permeability coefficient of the rutting board is generally the same, and the water-permeability coefficient tends to be zero. When the amount of HPBFA, SBS-modified emulsified asphalt, and the asphalt commonly used in fog-sealing layers is 0.6 kg/m^2^, the tank-plate specimens can be completely impermeable. When the amount of BE-4-emulsified asphalt is 0.8 kg/m^2^, the tank-plate specimens can be completely impervious to water.

When the total content of asphalt, SBS-modified emulsified asphalt, and HPBFA in the fog-sealing layer is ~0.6 kg/m^2^, the rut board can be completely water-free. When the BE-4-emulsified asphalt content is ~0.8 kg/m^2^, the specimen can be completely impermeable to water. This may be related to the contact angle and hydrophobicity of the material. Usually, when the contact angle is larger, better hydrophobicity and stronger permeability of the material are achieved.

#### 3.2.2. Bond Shear Strength

For three common pavement structures, namely, AC-13, SMA-13, and cement concrete, the test results of the bonding shear strength of HPBFA-cured specimens are shown in Figure 12. There were six parallel samples in a group, and the results were representative values according to the Grubbs method.

The cement concrete exhibits the lowest bond shear strength, while AC-13 and SMA-13 show similar bond shear strengths. Compared with BE-4-emulsified asphalt, SBS-modified emulsified asphalt, and the asphalt commonly used in fog-sealing layers, the bond shear strength of HPBFA-cured AC-13 specimen increases by 15.2%, 8.6%, and 4.1%, respectively, while HPBFA-cured SMA-13 specimens show corresponding improvements of 29%, 14%, and 8%, respectively, and HPBFA-cured cement concrete specimens increase by 21%, 17%, and 6%, respectively. This is because the fractured surface of the HPBFA-cured material is rough and shows high fracture toughness. Moreover, the surface of the cured material features a network skeleton structure, effectively improving the strength and wear resistance of HPBFA.

#### 3.2.3. Water Stability

Figure 13 shows the water stability test results of the specimens. There were six parallel samples in a group, and the results were representative values according to the Grubbs method.

Figure 13 shows that the MS_0_ and TSR values of the HPBFA specimens are improved compared with the control specimens. The MS_0_ values of the BE-4-emulsified asphalt, SBS-modified emulsified asphalt, the asphalt commonly used in fog-sealing layers, and HPBFA specimens increase by 5.1%, 7.0%, 9.6%, and 10.6%, respectively, while the TSR values increase by 6.8%, 13.6%, 13.6%, and 15.3%, respectively. These results indicate that HPBFA exhibits better permeability and cohesion and can better penetrate the pavement voids and bind loose aggregates.

#### 3.2.4. Aging Resistance

Table 7 presents the mass-loss rate of the specimens after photothermal aging. There were six parallel samples in a group, and the results were representative values according to the Grubbs method.

Table 7 shows that, compared with the BE-4-emulsified asphalt, SBS-modified emulsified asphalt, and the asphalt commonly used for fog-sealing layers, the HPBFA-cured material exhibits the lowest mass-loss rate after photothermal aging, which is reduced by 10%, 31%, and 40%, respectively. Hence, the aging performance of the HPBFA-cured material is good.

## 4. Microstructure and Strength Formation Mechanism

### 4.1. Micromorphology

The HPBFA-cured specimens were prepared, and the fracture surface and surface were etched with petroleum ether and sprayed with gold, respectively. A Quanta200 environmental scanning electron microscope (FEI Company, Eindhoven, Netherlands) was used to observe the section and surface appearance structure of the specimen (Figure 14). 

Based on Figure 14, the fracture surface of the HPBFA-cured material is rough and uneven, indicating good fracture toughness. After the specimen surface is treated with petroleum ether, the asphalt is etched, and the epoxy network framework is clearly visible, indicating that the epoxy system in the cured material (waterborne epoxy resin emulsion and curing agent) could form a continuous three-dimensional network structure [30]. All the phenomena indicate that the HPBFA-cured material shows good structural and mechanical properties and can effectively improve the strength and wear resistance of asphalt materials.

### 4.2. Hydrophobic

An OCA-20 video optical contact angle measuring instrument (DATAPHYSICS, Filderstadt, Germany) was used to measure the contact angle between the evaporation residue of emulsified asphalt and HPBFA-cured material and compare the sealing hydrophobicity (Figure 15). Figure 16 presents the test results.

SCA20 software was used to calculate the contact angle of the evaporation residue of emulsified asphalt, which is approximately 98.8°. The contact angle of the HPBFA-cured material is approximately 112.6°. Further, as observed in Figure 16, both the evaporation residue of emulsified asphalt and HPBFA-cured material feature hydrophobic surfaces. Moreover, the HPBFA-cured material shows a greater contact angle than the evaporation residues of emulsified asphalt, indicating that the epoxy-emulsified asphalt exhibits better hydrophobicity and excellent permeability when it is used as fog-sealing material.

### 4.3. Thermal Stability

A Pyris-I TGA instrument (PERKINELMER, Waltham, MA, USA) was used to conduct the thermal stability analysis of the HPBFA-cured material and matrix asphalt specimen, and the curve is shown in Figure 16. The flow rate of the nitrogen atmosphere was set to 50 thermogravimetric (TG) analysis mL/min, and the heating rate was set to 5 °C/min.

Based on Figure 17, the TG curves of the two materials show the same trend, and each exhibits a TG plateau. The initial thermal weight loss temperature of matrix asphalt is approximately 380 °C, and the asphalt undergoes complete decomposition or oxidization when the temperature is increased to 490 °C; hence, the temperature decomposition range is 0–110 °C. The initial thermal weight loss temperature of the HPBFA-cured material is approximately 300 °C, and the temperature decomposition range is 0–160 °C, which is higher than that of the matrix asphalt. Compared with the matrix asphalt, although the initial thermal weight loss temperature of the HPBFA-cured material is lower, it is still higher than the environmental aging temperature. The incorporation of an epoxy resin system increases the temperature decomposition range of the material and decelerates the asphalt oxidation or decomposition rate. Therefore, the HPBFA-cured material exhibits better thermal stability and can improve the aging resistance of road surfaces when it is used as a fog-sealing adhesive material.

## 5. Conclusions

In this study, a type of fog-sealing adhesive material with high permeability and bonding performance was fabricated, and its technical performance was evaluated. The strength formation mechanism of the fog-sealing adhesive material was revealed via microcosmic analysis. The main conclusions are summarized as follows:(1)Using laboratory tests, the formulation of epoxy-emulsified asphalt with high permeability and bonding performance is proposed as the adhesive material for preparing the fog-sealing layer. The recommended formula is 80% emulsified asphalt and 2:1–3:1 epoxy ratio.(2)The technical performance of the proposed HPBFA is verified using laboratory tests. Compared with the domestic traditional fog-sealing adhesive material, the recommended formula shows better permeability, bond shear strength, water stability, and aging resistance.(3)The microstructural properties of the HPBFA-cured material are analyzed. Results show that the epoxy system in the HPBFA-cured material can form a continuous three-dimensional network structure, which can improve the strength and wear resistance of the material. The cured material shows a good hydrophobic surface, large contact angle, and good permeability. The initial thermal weight loss temperature of the HPBFA-cured material is considerably higher than the environmental aging temperature. Moreover, the maximum temperature decomposition range is 0–160 °C, and the material shows good thermal stability.

Herein, we proposed a high-permeability and high-bonding fog-sealing adhesive material. In the future, we will study the influence of different factors on the environment and apply this material to the actual asphalt pavement engineering to further observe and evaluate the pavement performance and durability of fog-sealing surfaces.

## Figures and Tables

**Figure 1 materials-14-03599-f001:**
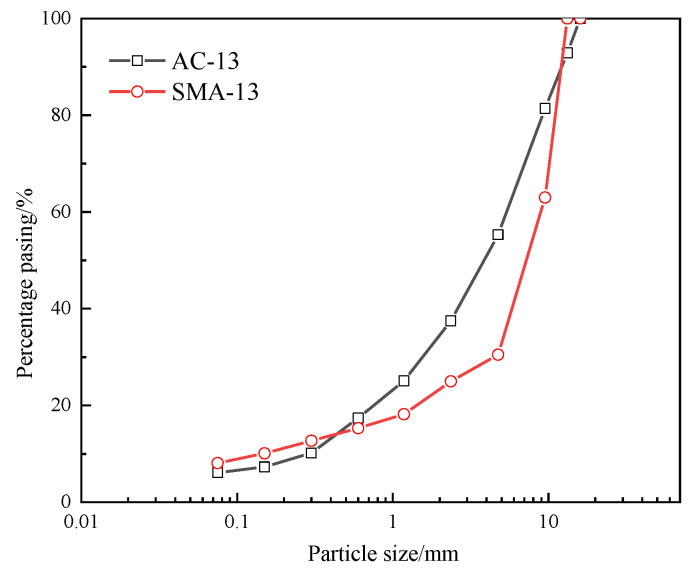
Gradation of the mixtures.

**Figure 2 materials-14-03599-f002:**
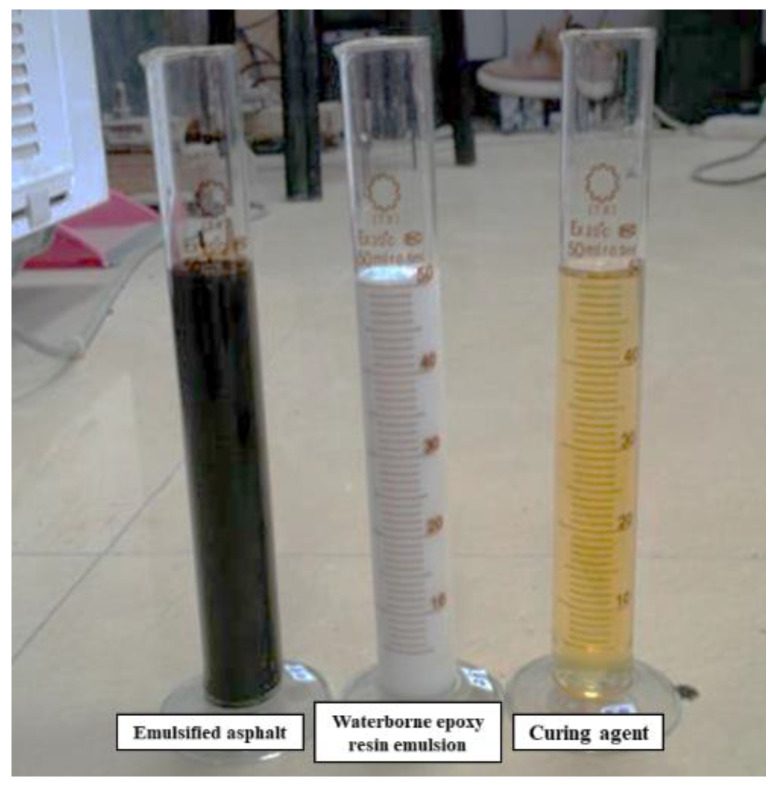
Three components of the epoxy-emulsified asphalt.

**Figure 3 materials-14-03599-f003:**
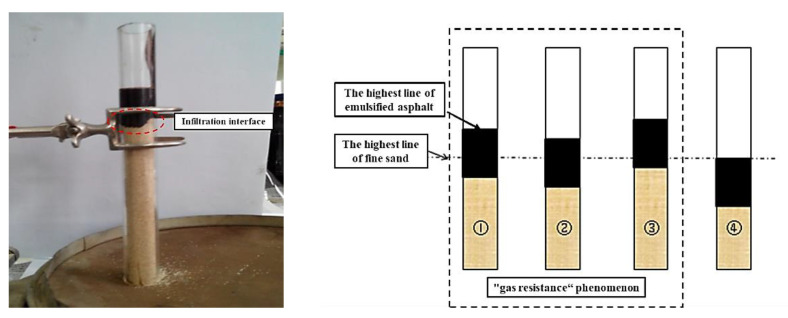
Problems in the existing penetration tests.

**Figure 4 materials-14-03599-f004:**
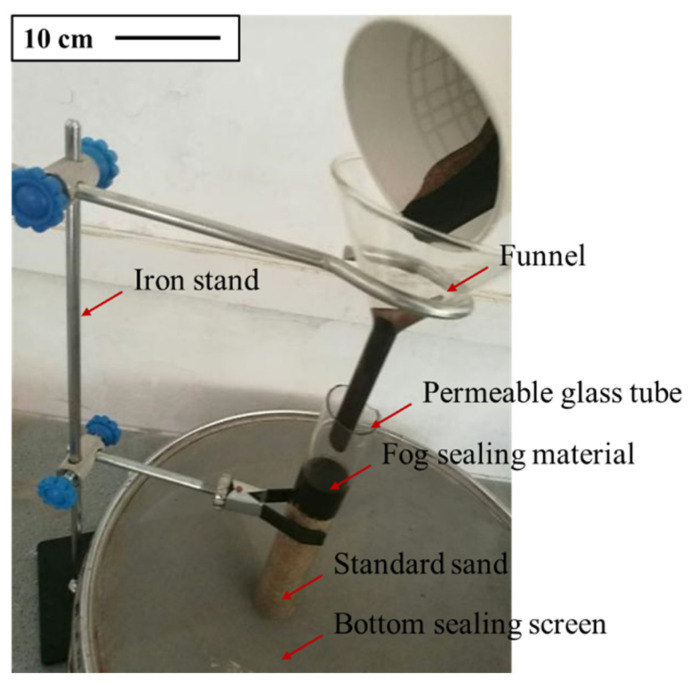
Improved penetration test device.

**Figure 5 materials-14-03599-f005:**
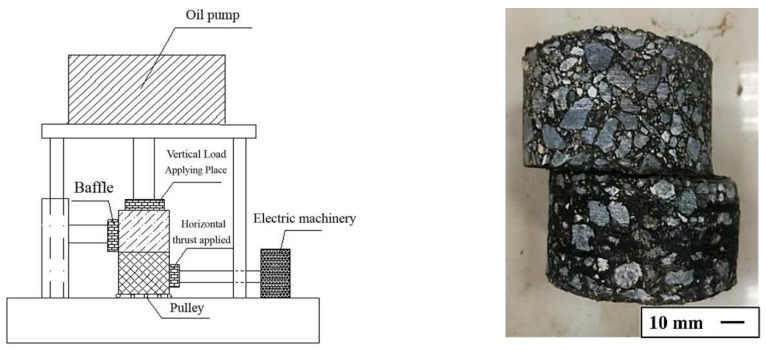
Schematic diagram of pavement interlayer material shear tester.

**Figure 6 materials-14-03599-f006:**
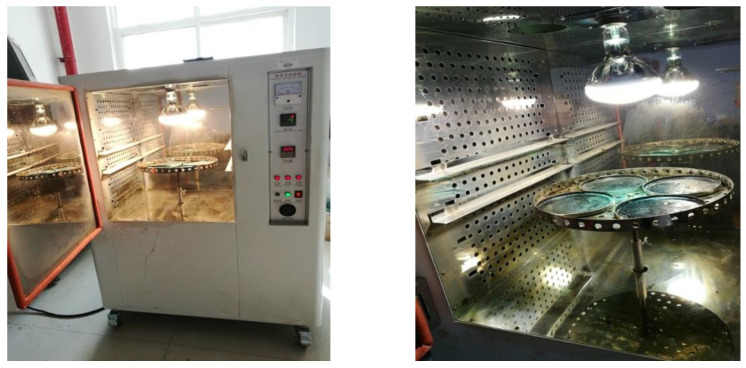
Antiyellowing aging box.

**Figure 7 materials-14-03599-f007:**
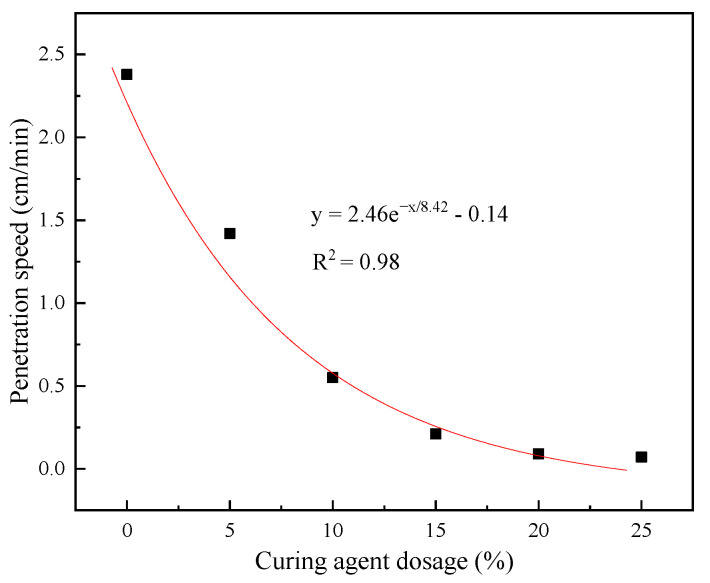
Relationship between the curing agent content and penetration velocity.

**Figure 8 materials-14-03599-f008:**
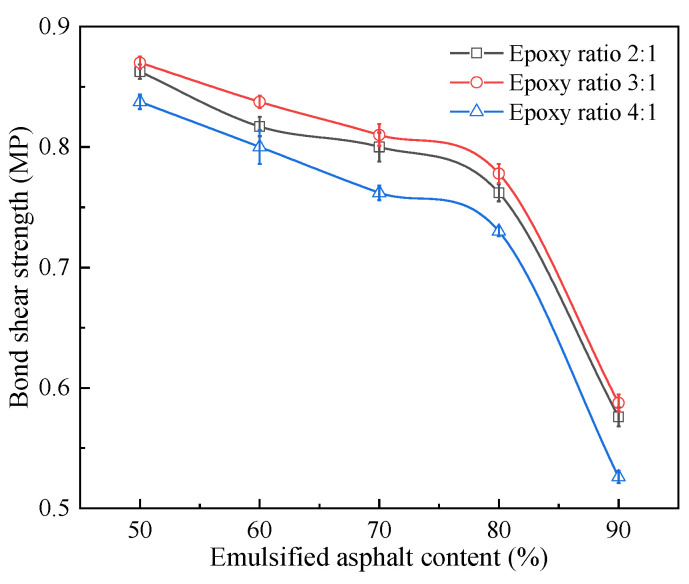
Bond shear strength test results.

**Figure 9 materials-14-03599-f009:**
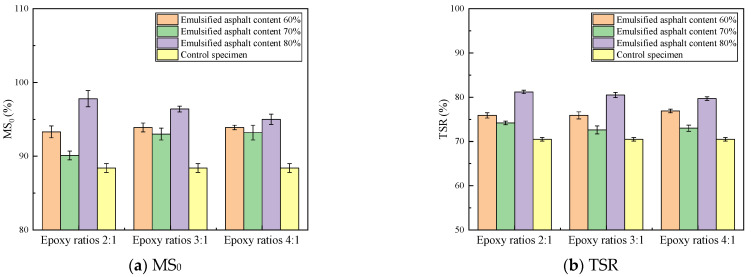
MS_0_ and TSR of specimens.

**Figure 10 materials-14-03599-f010:**
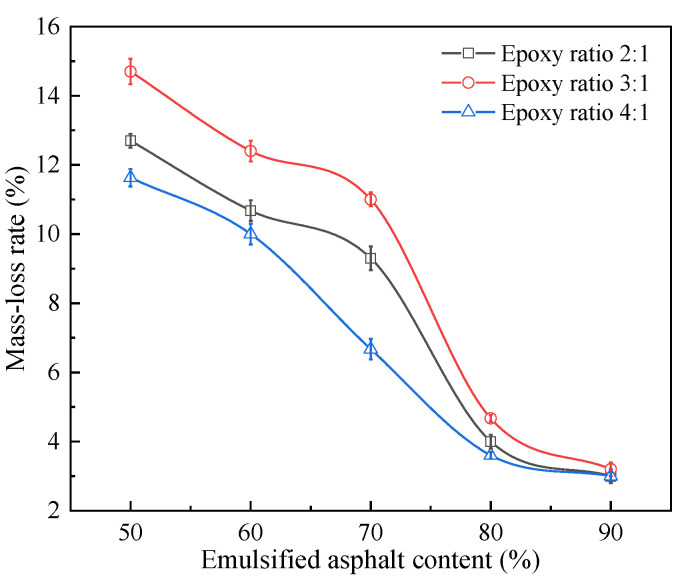
Mass-loss rate of epoxy-emulsified asphalt after photothermal aging.

**Figure 11 materials-14-03599-f011:**
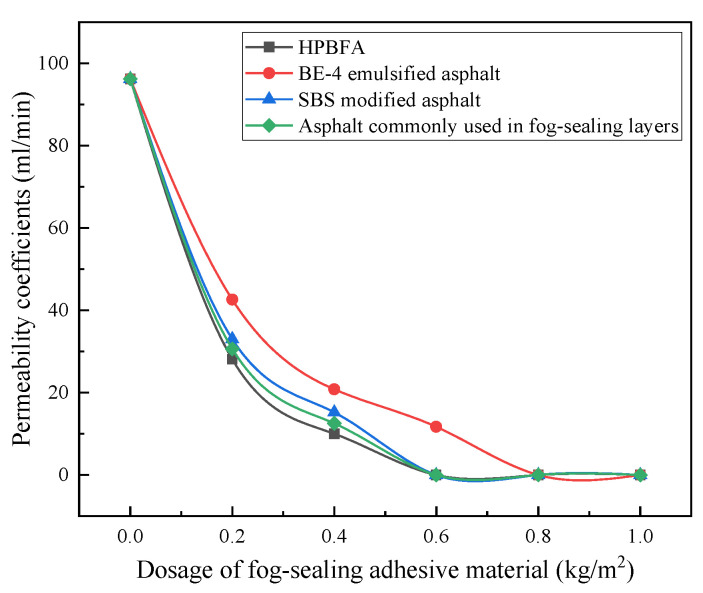
Experimental water-permeability coefficients.

**Figure 12 materials-14-03599-f012:**
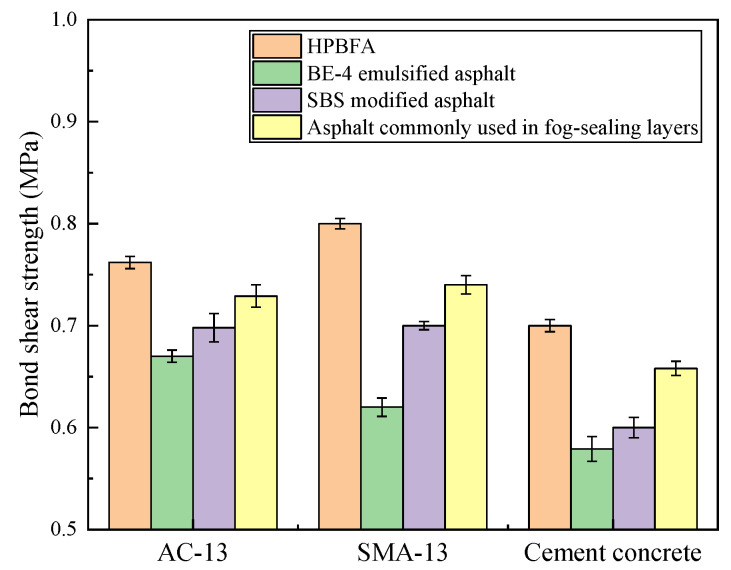
Bonding properties of various fog-sealing adhesive materials.

**Figure 13 materials-14-03599-f013:**
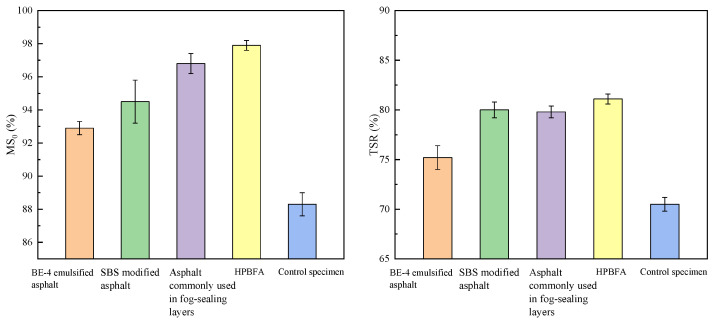
Results of water stability tests on different types of fog-sealing adhesive materials.

**Figure 14 materials-14-03599-f014:**
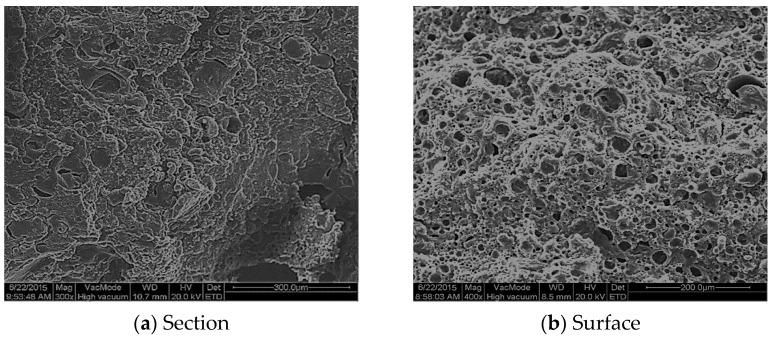
Micromorphology of HPBFA-cured specimens.

**Figure 15 materials-14-03599-f015:**
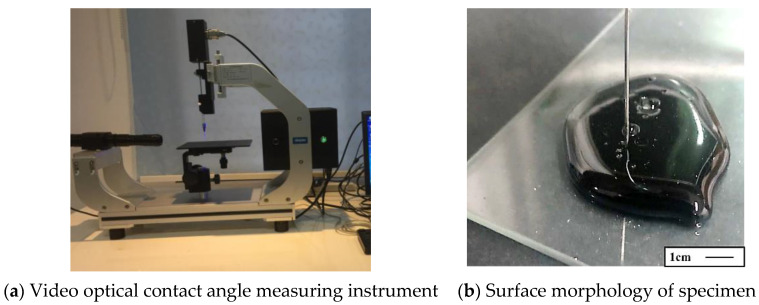
Contact angle test.

**Figure 16 materials-14-03599-f016:**
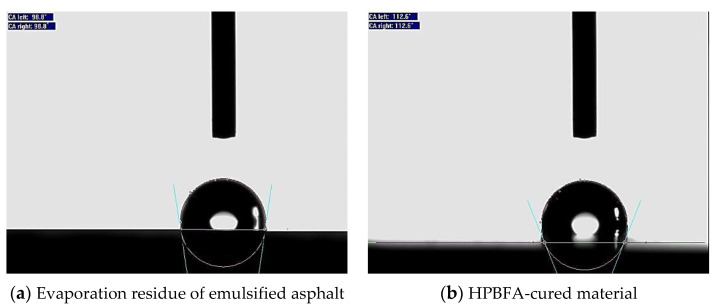
Contact angle of the cured material.

**Figure 17 materials-14-03599-f017:**
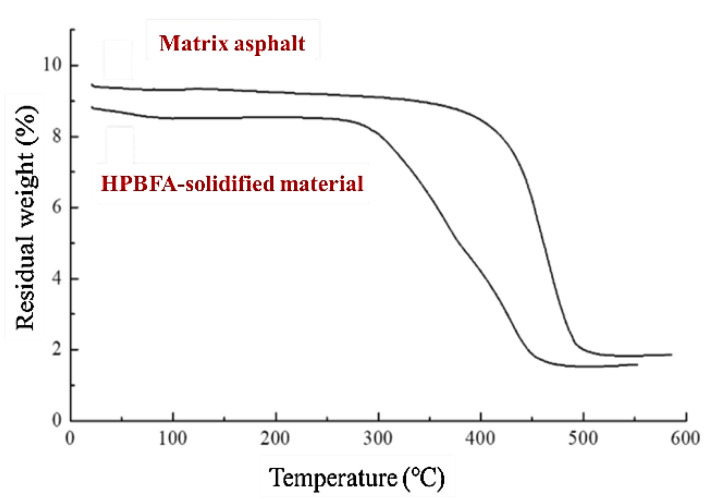
TG curve.

**Table 1 materials-14-03599-t001:** Technical properties of BE-4-emulsified asphalt.

Test Indicators		Tested Values	Standards
Demulsification speed		Slow crack	Slow crack
Particle charge		+	Cation (+)
Residual amount on sieve (1.18 mm sieve) (%)		0.01	≯0.1
Engla viscometer E25		3.5	1–6
Evaporative residues	Residual content (%)	50.7	≮50
	Penetration (100 g, 25 °C, 5 s) (0.1 mm)	88	50–300
	Ductility (15 °C) (cm)	>100	≮40
Storage stability at room temperature	1 d (%)	0.3	≯1
	5 d (%)	2.8	≯5

**Table 2 materials-14-03599-t002:** Technical properties of E-44 epoxy resin.

Softening Point (℃)	Epoxy Equivalent	Shrinkage Rate (%)	Coefficient of Thermal Expansion (%)	Epoxy Value (Eq/100 g)	Organochlorine (%)	Inorganic Chlorine (ppm)	Volatile (%)
16–24	212.6–238.1	<2	6–10.2	0.42–0.47	≯0.5	≯200	1.0

**Table 3 materials-14-03599-t003:** Technical properties of the curing agent.

Test Indicators	Active Ingredient Content (%)	Density (g/cm^3^)	Viscosity (mPa·s)	Active Hydrogen Equivalent
Test values	50.0 ± 1.0	1.05–1.12	5000–20,000	287

**Table 4 materials-14-03599-t004:** Permeability of different tested liquids.

Test Indicators	Emulsified Asphalt	Waterborne Epoxy Resin Emulsion	Curing Agent	Water
Residual sand quality (g)	49.3	49.2	57.9	49.1
Osmotic time (min)	0.69	0.72	>30	0.13
Penetration velocity (cm/min)	2.38	2.30	/	12.99

**Table 5 materials-14-03599-t005:** Recommended composition contents for preparing epoxy-emulsified asphalt.

Test Indicators	Recommended Contents for Following Components		
	Emulsified Asphalt	Curing Agent	Epoxy Ratio
Permeability	-	≯10%	-
Bond shear strength	≯80%	-	2:1–3:1
Water stability	80%	-	2:1–4:1
Aging resistance	≮80%	-	2:1–4:1

**Table 6 materials-14-03599-t006:** Results of permeability tests of different fog-sealing adhesive materials.

Test Indicators	HPBFA	BE-4-Emulsified Asphalt	SBS-Modified Emulsified Asphalt	Asphalt Commonly Used in Fog-Sealing Layers
Residual sand mass (g)	50.4	49.3	53.2	55.5
Penetration time (min)	1.14	0.69	6.52	1.97
Seepage velocity (cm/min)	1.29	2.38	0.16	0.35

**Table 7 materials-14-03599-t007:** Mass-loss rate of the specimens after photothermal aging.

Type of Fog-Sealing Adhesive Material	HPBFA	BE-4-Emulsified Asphalt	SBS-Modified Emulsified Asphalt	Asphalt Commonly Used in Fog-Sealing Layers
Mass-loss rate (%)	3.93	4.36	5.70	6.55

## Data Availability

The data used to support the findings of this study are included within the article.

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
