# Peer review of "Development and Performance Evaluation of a High-Permeability and High-Bonding Fog-Sealing Adhesive Material"

_materials, 2021, doi:10.3390/ma14133599_

Round 1
Reviewer 1 Report
I would like to thank you the authors for presenting the research data and preparing the article.
I found the paper to be interesting, research oriented and may be useful in engineering application.
However, in my opinion, for this to become a fact, it requires additions and clarifications.
The type of research presented is interesting, the range of research applied is wide and adequate to the objectives put forward but still the chapters Materials and Methods and Results and Discussion need to be extended and supplemented.
A research paper that is intended to be an implementation requires many iterations of tests and statistical analyses of test results. This is the aspect that was missing from the article for me. It was also missing in the conclusion. And the conclusion is positive in its justification but insufficient.
Can the authors address this comment and add in the text the number of tests performed, repetitions, statistical analyses if any, or justify the lack of them. This applies to each of the individual studies.
Please also include this in Conclusion chapter.
Environmental Impact is not mentioned at all but a future engineering application requires it.
Abrasiveness, suspended dust, predicted magnitude and extent of dispersion, safety to humans and the environment. Please address this issue.
On one hand proposed future application and on the other hand lack of statistical analysis and plans to substantiate the research results. Promising topic but research results currently unconfirmed.
Please be absolutely precise and refer to it in the publication.
Figure 17: No English description of the axes.
Reviewer 2 Report
The development of fog-sealing adhesives is of importance in the road industry.
The proposed manuscript presents the development and performance evaluation of a high-permeability and high-bonding fog-sealing adhesive material. The effects of the contents of emulsified asphalt, waterbone epoxy resin emulsion and curing agent are discussed.
The manuscript is composed of an introduction giving a fair background on the subject and presenting the novelty of the research, a Materials and Methods section which would benefit additional information, a Results and Discussion section, a brief fourth section called ‘Microstructure and Strength Formation Mechanism’ and a last Conclusion section which summarizes well the different outputs. The article may be of interest of the readers of the journal and more generally the civil engineering community.
Some points should however be precised before an eventual publication:
- abstract: define HPFA
- l 92: please add some sentences presenting the Table with the main information
- l 95: please provide some detail about the emulsification procedure
- Table 4: please consider changing the Table into PSD curves
- l 113: detail the mixture procedure
- Figure 4: the figure is not useful in my opinion
- Figure 5 and fig 6: the two figures might be merged in fig 5 a) and fig 5 b)
- l 188: precise 50.05 mm value as the diameter is 100 mm (l 173)
- l 393: precise Pyris II is a TGA instrument
Reviewer 3 Report
I would like to thank the authors for their contribution and presentation of the results of such an interesting topic. However, in the course of writing, the authors did not avoid fine mistakes and mistakes that require explanation. They are listed numerically below:
1) Please explain why some of the standard values in Table 1 are not met for the tested mixtures?
2) Please add the SDS od curing agent and epoxy resin
3)On figures 3, 4 and 6please add the scale bar.
4) Please compare your results about bond strength and water stability with other authors. Because on this moment you write only research raport.
5) On figures 9 and 11 please extend the x-axis, because some points were cut off.
6) Please cut the ( Waterborne epoxy resin emulsion column) because you do not write any information about it.
7) Figure 15 the photo is too contrasting, please change the intensity to show the structure artifacts accurately.
8) Figure 16 please add scale bar and photos after the wetting test. How does a drop of liquid behave on the tested surfaces?
9) Figure 17 please use a European letter on the graphs.
In my opinion, the authors lack a number of comparisons to the currently used asphalt mix additives. Authors should focus on comparing their results to other scientists, and this should be included in the "Results and Discussion" chapter. Currently, a very narrow and low-volume discussion lowers the scientific level of research, which is carried out very broadly and in accordance with standards. If the authors complete the missing comparisons, the article will be complete. As it stands, it looks more like a research report than a scientific endeavor.
In addition, the authors in their research did not include in any way the calculations related to the measurement error of key parameters for asphalt mixtures. The charts should include the extension of this information, e.g. in the form of measurement errors bar, or tables under the graph about standard or calculation errors.
Reviewer 4 Report
The manuscript “Development and performance evaluation of a high-permeability and high-bonding fog-sealing adhesive material” studied the effects of the contents of emulsified asphalt, waterborne epoxy resin emulsion, and curing agent on the permeability, bond shear strength, water stability, and aging resistance of epoxy-emulsified asphalt. This manuscript is well written and well structured, with a proper English grammar and syntax. From a scientific point of view the manuscript shows a high degree of soundness, supported by pertinent references. To conclude, I suggest this manuscript to be published in the journal “Materials” after the below minor revisions:
- It would be convenient to include a table of abbreviations.
- Please provide the manufacturers for all the equipment used.
- The final question concerns for the economically aspects of the process?
- The authors should provide a better and detail description about the emulsification procedure (line 95), as well as a detail description for the mixture procedure (line 113).
Round 2
Reviewer 1 Report
I thank the authors for their efforts in improving the publication. I still have some doubts about the scientific level of the publication, but as I mentioned earlier I consider the publication to be valuable.
Reviewer 3 Report
Thank You for all corrections. In this moment manuscript is improved. The authors did a great deal of work and corrected all the bad sides of the manuscript. I accept the correction. Thank You very much.